# An Automated Cartridge-Based Microfluidic System for Real-Time Quantification of *BCR::ABL1* Transcripts in Chronic Myeloid Leukemia: An Italian Experience

**DOI:** 10.3390/ijms26188932

**Published:** 2025-09-13

**Authors:** Alice Costanza Danzero, Enrico Marco Gottardi, Fabrizio Quarantelli, Ciro Del Prete, Alessandra Potenza, Claudia Venturi, Paola Berchialla, Francesca Guerrini, Clara Bono, Emanuela Ottaviani, Sara Galimberti, Carmen Fava, Barbara Izzo

**Affiliations:** 1Department of Clinical and Biological Sciences, Turin University, 10043 Orbassano, Italy; enricogottardi@libero.it (E.M.G.); paola.berchialla@unito.it (P.B.); carmen.fava@unito.it (C.F.); 2Oncologic Hematology and Cytogenetics Laboratory, CEINGE Advanced Biotechnology “Franco Salvatore”, 80131 Naples, Italy; quarantelli@ceinge.unina.it (F.Q.); delpreteci@ceinge.unina.it (C.D.P.); potenza@ceinge.unina.it (A.P.); 3“Seràgnoli” Institute of Hematology, IRCCS University Hospital of Bologna, 40138 Bologna, Italy; ventclod85@gmail.com (C.V.); emanuela.ottaviani@aosp.bo.it (E.O.); 4Department of Clinical and Experimental Medicine, Hematology, University of Pisa, 56126 Pisa, Italy; guerrinifra@libero.it (F.G.); c.bono3@studenti.unipi.it (C.B.); sara.galimberti@unipi.it (S.G.); 5Department of Molecular Medicine and Medical Biotechnology, University of Naples “Federico II” and CEINGE Advanced Biotechnology “Franco Salvatore”, 80131 Naples, Italy

**Keywords:** chronic myeloid leukemia, *BCR::ABL1*, molecular response monitoring, Xpert® BCR-ABL Ultra

## Abstract

Chronic myeloid leukemia (CML) is a clonal myeloproliferative disorder caused by the *BCR::ABL1* fusion gene, resulting from a reciprocal translocation between chromosomes 22 and 9. Quantification of *BCR::ABL1* transcript levels in peripheral blood by RT-qPCR represents the gold standard for molecular response (MR) monitoring, providing essential clinical information on treatment efficacy. Xpert^®^ BCR-ABL Ultra is a fully automated in vitro diagnostic test that quantitatively detects e13a2 and e14a2 *BCR::ABL1* transcripts using a single-use cartridge that integrates RNA extraction, cDNA synthesis, nested real-time PCR, and signal detection within a rapid, closed, and user-friendly system. In this study, we evaluated Xpert^®^ BCR-ABL Ultra as an alternative to validated systems currently used by four highly specialized Italian laboratories affiliated with the Italian national laboratory network for CML. A total of 129 peripheral blood samples from CML patients at various disease stages, along with two external quality control materials, were analyzed. We assessed the test’s repeatability, specificity, and stability. Concordance of *BCR::ABL1*%IS values generated by the different methods was evaluated using EUTOS criteria and Bland–Altman analysis. Finally, MR value concordance was analyzed based on European LeukemiaNet recommendations or calculated using the formula 2 − log_10_(*BCR::ABL1*%IS). Xpert^®^ BCR-ABL Ultra demonstrated high repeatability and stability. The *BCR::ABL1*%IS values obtained with this assay showed strong concordance with those generated by local reference methods, and MR classifications were consistent across platforms. These findings confirm the robustness, accuracy, and efficiency of the Xpert^®^ BCR-ABL Ultra assay, supporting its use as a reliable alternative to currently validated systems for the routine clinical monitoring of CML patients.

## 1. Introduction

Chronic myeloid leukemia (CML) accounts for approximately 15% of all leukemia cases. It is characterized by the presence of the Philadelphia chromosome that arises from a reciprocal translocation between chromosomes 9 and 22 [t(9;22)] and generates the *BCR::ABL1* fusion gene [1]. This gene fusion typically involves *BCR* exon 13 or 14 and *ABL1* exon 2, giving rise to the e13a2 (38%) and e14a2 (62%) transcripts, respectively. These transcripts encode the p210 protein, which exhibits constitutively active tyrosine kinase activity. Less commonly, alternative fusion transcripts such as e1a2 and e19a2 are detected, leading to the production of p190 and p230 proteins, respectively. These variants are observed in approximately 2% of CML cases, including during blast phase progression [2].

Quantification of *BCR::ABL1* transcript levels in peripheral blood is the gold standard for managing CML patients undergoing tyrosine kinase inhibitor (TKI) therapy. Reverse transcription quantitative polymerase chain reaction (RT-qPCR), calibrated to the World Health Organization (WHO) International Standard, is the preferred method for monitoring the response to TKIs. Molecular monitoring is recommended at least every three months until major molecular remission (MMR) is reached and confirmed. Once stable MMR or deeper molecular responses are achieved, testing can be performed at intervals of four to six months. More frequent assessments are warranted if transcript levels show variability or increase, or when evaluating eligibility for treatment discontinuation and during subsequent follow-up in selected patients [3].

PCR assays with a sensitivity of at least a 4.5 log reduction from baseline are recommended for accurately measuring *BCR::ABL1* transcript levels. A standardized RT-qPCR protocol includes high-quality RNA extraction and optimization, followed by reverse transcription to cDNA and subsequent amplification, detection, and accurate quantification [4,5]. Although highly sensitive and reliable, this multi-step process is time consuming and technically demanding, with potential sources of error and variability at each stage, from RNA extraction to data interpretation. These challenges highlight the need for more streamlined approaches that maintain analytical accuracy while reducing complexity.

Xpert^®^ BCR-ABL Ultra (Cepheid, Sunnyvale, CA, USA) is an in vitro diagnostic test for the quantification of *BCR::ABL1* and *ABL1* mRNA transcripts in peripheral blood specimens of diagnosed t(9;22) positive CML patients. The Xpert^®^ BCR-ABL Ultra test is based on a single-use cartridge system that fully automates the quantitative process, integrating RNA extraction, reverse transcription, amplification, and detection within a closed system in under three hours. This innovative approach simplifies the workflow and minimizes the risk of contamination [6], making Xpert^®^ BCR-ABL Ultra a robust, rapid, and efficient tool for molecular diagnostics.

Over the past 15 years, substantial efforts have been made worldwide to harmonize molecular methods for *BCR::ABL1* quantification, particularly through the development and implementation of the International Scale (IS), which enables consistent interpretation of molecular responses across laboratories. A fundamental component of this harmonization process was the introduction of laboratory-specific conversion factors (CFs), developed to align individual assay results with the IS and minimize inter-laboratory variability [7]. In this context, Dominy et al. demonstrated that the Xpert^®^ BCR-ABL Ultra test is effective in confirming laboratory-specific CFs for molecular response (MR) monitoring, offering a rapid and reliable approach to assess assay performance and ensure result comparability [8].

Building on this evidence, our study broadens the validation of Xpert^®^ BCR-ABL Ultra by comparing it across four different methods in a multi-center Italian setting, using real-world patient samples, external quality controls, and stability testing to reflect routine clinical practice. By focusing on multiple centers and real-world applicability, this work advances beyond prior validations, providing robust evidence of assay performance under different conditions. The four laboratories participating in the study are part of the Italian national laboratory network for CML, which drafts and distributes the R.I.L. (Recommendations and Laboratory Indications) and ensures that participating centers report results according to the International Scale (IS) through yearly External Quality Assessment (EQA) rounds, thereby maintaining method standardization. Within this framework, introducing new tools capable of improving laboratory performance remains a key area of research and technical development.

The Xpert^®^ system thus represents a standardized and automated alternative that could reduce inter-laboratory variability and improve clinical decision-making through accurate molecular monitoring. To assess its performance, we compared Xpert^®^ BCR-ABL Ultra with established RT-qPCR reference methods using EUTOS criteria [9], evaluated the stability of its results on blood samples that were analyzed immediately and 24 h after collection, and tested its accuracy and reliability using two external quality control materials. Offering a fast, automated, and standardized approach, the Xpert^®^ system has the potential to enhance the reliability and efficiency of molecular monitoring in routine clinical practice.

## 2. Results

### 2.1. Repeatability Assessment

A total of 129 fresh peripheral blood samples from CML patients were collected: 41 samples were collected by lab #1, 33 samples by lab #2, 31 samples by lab #3, and 24 samples by lab #4.

Repeatability was assessed in duplicate using the local system and in triplicate using the Xpert^®^ BCR-ABL Ultra assay (Cepheid method). The coefficient of repeatability was calculated as the upper prediction limit of the absolute difference between two measurements obtained with the same method [10]. Given that the samples had different MR levels, the standard deviation (SD) of replicate measurements tended to increase with higher transcript levels. To account for this variability, we calculated the coefficient of variation (CV), which expresses the SD relative to the mean value (Table 1).

### 2.2. Concordance Assessment: EUTOS Criteria

Table 2 shows the percentage of the 107 positive quantitative results, stratified by laboratory, for which the ratio of Xpert^®^ BCR-ABL Ultra results to comparator results fell within the ranges specified by the EUTOS criteria [9].

### 2.3. Concordance Assessment: Bland–Altman Analysis

Bland–Altman analysis was performed to assess the bias between the data obtained with Xpert^®^ BCR-ABL Ultra and those obtained using local methods (Table 3). In the Bland–Altman plot, there were three points that exceeded the 95% LoA, which represents a 10% outlier rate. This primarily reflects the expected higher variability at low MR levels (MR^1^ and MR^2^), where relative measurement differences are inherently larger. Importantly, these levels correspond to a high disease burden. Additionally, the numerical differences between MR^1^ and MR^2^ do not affect clinical decision-making since they are typically reported together as <MR^3^ on the patient report. Therefore, these outliers do not indicate a systematic bias of the Xpert^®^ BCR-ABL Ultra assay and are consistent with the variability encountered in real-world laboratory settings. Appendix A shows the boxplot of the value distribution, stratified by MR.

### 2.4. Concordance Assessment: MR Analysis

Table 4 and 5 summarize the distribution of samples across molecular response (MR) categories and the corresponding concordance rates. For each of the 129 samples, MR values were assigned using two different methods: the European LeukemiaNet (ELN) recommendations [4] (Table 4) and a mathematical formula for calculating the logarithmic reduction (Table 5). This dual approach allowed us to evaluate the concordance between local methods and the Xpert^®^ BCR-ABL Ultra assay (Cepheid method) using two complementary strategies for MR classification.

### 2.5. Stability Assessment and Test on Healthy Donors

Another objective was to simulate real-life conditions, so we evaluated the stability of the test 24 h after sampling. We selected 51 samples, which were analyzed upon arrival (Time 0) and after 24 h (Time +24 h). In total, 23 samples were stored as lysates at −20 °C, 14 samples were stored as whole blood at room temperature, and 14 samples were stored as whole blood at 4 °C to simulate three conditions that may occur in routine workflows of hematology–oncology diagnostic laboratories. For this comparison, the mean and standard deviation (SD) of MR values at Time 0 and at +24 h were calculated for each storage condition, along with the % deviation from Time 0. The overall % deviation from Time 0 was minimal and varied by storage condition: lysates stored at −20 °C showed a deviation of +0.47%, whole blood stored at room temperature (RT) showed a deviation of −5.55%, and whole blood stored at 4 °C showed a deviation of −1.78%. These data are shown in Appendix A.

Additionally, five healthy donors from each laboratory, yielding a total of 20 samples, were evaluated using both methods. No false positive samples were identified.

### 2.6. Tests on External Quality Control Materials

In each laboratory, five samples of lyophilized cell line from the AcroMetrix™ BCR-ABL Panel and two lyophilized samples from the UK NEQAS control panel, with known titers, were tested on the Cepheid cartridges (Appendix A). We analyzed the data from the four laboratories all together because there were a low number of repetitions and the cartridges used in the different centers were from the same lot.

The average values obtained by the four laboratories demonstrated high comparability to the reference values of the analyzed materials, indicating consistent and reliable performance across different sites. Measurements on the AcroMetrix™ reference materials using Xpert^®^ BCR-ABL Ultra showed deviations from expected values ranging from -27.2% to +52.9%. Excellent agreement was observed at clinically relevant low levels (MR^4^–MR^4.5^), whereas higher deviations occurred at intermediate (MR^3^) and high (MR^1^) levels. Measurements on UK NEQAS reference materials using Xpert^®^ BCR-ABL Ultra showed deviations from expected values of -33.7% and -7.5%.

The decision to test these two reference materials was driven not only by the need to verify the results obtained from the four different laboratories, but also by the intention to confirm their suitability for use on the Xpert^®^ BCR-ABL Ultra assay in periodic external quality control rounds that are already being conducted at the international level.

## 3. Discussion

The main objective of this study was to demonstrate that the Xpert^®^ BCR-ABL Ultra assay produces results consistent with those obtained using methods routinely employed in highly specialized laboratories.

We compared different approaches using 129 leftover peripheral blood samples from CML patients with varying *BCR::ABL1/ABL1* transcript levels, ranging from undetectable to 10%. Measurements were performed simultaneously in four Italian laboratories using Xpert^®^ BCR-ABL Ultra on the GeneXpert^®^ System and analyzed with the GeneXpert^®^ software. These data were then compared with results obtained using the methods recommended in the R.I.L. (Recommendations and Laboratory Indications) published and disseminated by the Italian national laboratory network for CML. The results, derived from data stratified by method and molecular response (MR) class, showed good repeatability for both the local and Cepheid methods. Specifically, the coefficient of variation (CV) for nearly all measurements was below 1 (Table 1), indicating low variability across replicates. Furthermore, the coefficient of repeatability was small relative to mean measurement values for nearly all sample groups. These findings suggest that variability between repeated measurements is minimal, allowing us to use the mean of replicates as a reliable representative value for subsequent method comparisons.

Acceptable concordance is defined by meeting at least two out of the three EUTOS criteria [9]: at least 50% of samples within a 2-fold range, at least 75% within a 3-fold range, and at least 90% within a 5-fold range. In this study, the two methods can be considered concordant as all three EUTOS criteria were satisfied. Furthermore, a more detailed analysis showed that the three EUTOS criteria were also met within each individual laboratory, with the exception of the first criterion in lab #4 (Table 2).

The Bland–Altman analysis, stratified by MR (Table 3), showed that, as expected, the variability of differences was not uniform across the measurement ranges used to define MR. Greater bias and wider limits of agreement were observed in MR^1^ and MR^2^, which correspond to *BCR::ABL1*%IS values above 0.1%. These classes are typically grouped and reported as <MR^3^, reflecting a high disease burden. In this case, fine measurement precision is not clinically critical. In contrast, from MR^4^ onwards—corresponding to *BCR::ABL1*%IS values of ≤0.01%—the need for analytical precision becomes increasingly important [11]. In these categories, particularly MR^4.5^ and MR^5^ (≤0.0032% and ≤0.001%, respectively), both bias and variability were minimal. The 95% limits of agreement did not exceed 0.01% for MR^4^ and 0.0032% for MR^4.5^ and MR^5^. These findings demonstrate a strong agreement between the two methods at deep molecular response levels, where analytical sensitivity and precision are essential for reliable clinical decision-making [12].

Given the clinical and prognostic importance of accurate MR assessment for patient management, we further analyzed the concordance between the Cepheid and local methods in assigning MR values to the analyzed samples. This comparison was conducted using two distinct calculation approaches.

The MR values obtained by applying the European LeukemiaNet (ELN) recommendations [4] showed varying degrees of concordance when samples were stratified by MR class, with lower concordance observed for MR^3^ and MR^4^. Overall concordance was 71.3%. MR values assigned by the Xpert^®^ BCR-ABL Ultra assay tended to be higher than those obtained using local methods (Table 4). In the 36 discordant cases, where local MR values were lower than those obtained with Xpert^®^ BCR-ABL Ultra, the discrepancy is most likely attributable to methodological differences rather than a genuine difference in analytical sensitivity. A key contributor may be differences in *ABL1* quantification, which tend to produce systematically higher MR values with the Cepheid method. In some cases, this resulted in samples being classified just above clinically relevant thresholds such as MR^3^, potentially affecting treatment response categorization under ELN guidelines. This underscores the need for consistent methodology when MR values are near decision boundaries, and for further investigation into the clinical relevance of MR values obtained by applying ELN criteria to Cepheid results. Such an approach differs from the assay’s intended design, which generates MR values through an automated mathematical calculation within the instrument.

A second approach for evaluating MR concordance involved assigning molecular response values using the Cepheid-based calculation. The resulting MR values were grouped into four predefined categories: <MR^3^, MR^3–3.99^, MR^4–4.49^, and >MR^4.5^. Results were considered concordant when the difference between the MR values calculated using the local method and those obtained with Xpert^®^ BCR-ABL Ultra did not exceed 0.5 log. Compared with the analysis based on ELN guidelines, this approach yielded equal or higher concordance rates in most categories, with a particularly notable improvement in the MR^4–4.49^ range (from 33.3% to 96.3%). Overall concordance rose from 71.3% to 89.9%, and most discordant MR values were higher when measured using the Cepheid method (Table 5).

The Xpert^®^ BCR-ABL Ultra assay does not report the *ABL1* copy number in its output. Rather, it only provides a result if the *ABL1* copy number exceeds 32,000. Consequently, if ELN recommendations are strictly applied, no sample analyzed with the Cepheid method can technically be classified as MR^5^, as the guidelines require more than 100,000 *ABL1* copies to confirm this level. It should be noted, however, that the Cepheid method does quantify the *ABL1* copy number, and this value directly influences the reported *BCR::ABL1*%IS and the calculated MR value. Therefore, when MR is calculated using the formula implemented by the Xpert^®^ BCR-ABL Ultra assay, higher MR values can be achieved, and concordance with local methods is substantially improved.

The results of the measurements performed on external quality control materials using the Xpert^®^ BCR-ABL Ultra assay (Cepheid method) are consistent with the reference values provided by the manufacturer (Appendix A). Although further statistical analyses were limited due to the small dataset, these findings support the suitability of the Xpert^®^ BCR-ABL Ultra assay for use in external quality control rounds.

To simulate real-world conditions, we assessed the stability of the test results 24 h after sample collection. The analysis of 51 samples that were tested immediately and again after 24 h demonstrated overall excellent stability. Concordance across different storage conditions—whole blood at room temperature (RT) or 4 °C, and lysate at −20 °C—indicates that these conditions did not meaningfully affect the results (Appendix A). The largest deviation was observed in samples stored at RT, which is the condition most prone to RNA degradation. Despite the limited sample size and the need for further confirmation, these results support the robustness of the Xpert^®^ BCR-ABL Ultra assay, making it suitable for integration into typical clinical laboratory workflows where processing delays beyond 24 h from collection may occasionally occur. Furthermore, all healthy control samples tested negative across both local and Cepheid methods, confirming the specificity and reliability of the Xpert^®^ BCR-ABL Ultra assay.

Overall, this study demonstrates that the Xpert^®^ BCR-ABL Ultra assay yields results comparable to those obtained with methods currently employed in routine clinical practice. Its robustness was confirmed through comparison with four different platforms used across four Italian laboratories that are highly specialized in onco-hematology diagnostics. We therefore conclude that the Xpert^®^ BCR-ABL Ultra test is a rapid, robust, and efficient platform that significantly simplifies laboratory workflows and minimizes the risk of error. It should be considered a valid alternative among the systems validated and used by the laboratories belonging to the Italian national laboratory network for CML.

## 4. Material and Methods

### 4.1. Cohort of Patients and Samples Collection

We used leftover peripheral blood (PB) in EDTA tubes collected from CML patients for routine molecular *BCR::ABL1* measurements. Regarding healthy subjects, we harvested 5 mL of PB in EDTA tubes from scientists involved in this research, following the institution’s guidelines [13]. Each participant gave his/her consent to sampling.

A total of 149 CML blood samples were acquired and tested in parallel at four Italian laboratories located in the hematology departments of Bologna (lab #1), Napoli (lab #2), Pisa (lab #3), and Orbassano (Turin) (lab #4).

The total collected samples had the following transcript levels:66 samples detectable from MR^3^ (0.1% IS) to MR^1^ (10% IS);27 samples detectable at MR^4^ (0.01% IS);16 samples detectable at MR^4.5^ (0.0032% IS);20 samples undetectable or detectable at MR^5^ (<0.001% IS);20 samples from healthy subjects.

The number of samples at each molecular response level was inherently limited by the availability of the leftover diagnostic material. Notably, there were fewer samples at MR^4.5^ and MR^5^, reflecting the lower frequency of patients at these deep molecular response levels in routine laboratory practice.

All collected samples were analyzed immediately upon arrival (Time 0). In addition, 51 of these samples were re-tested with Xpert^®^ BCR-ABL Ultra after 24 h (Time +24 h) following storage under different conditions. Specifically, whole blood was stored at room temperature (RT) or at 4 °C, and lysate was stored at −20 °C.

### 4.2. Xpert^®^ BCR-ABL Ultra Analysis

Each sample was analyzed in triplicate, with 3–4 mL of the same PB loaded onto each of the three Xpert^®^ BCR-ABL Ultra cartridges (Lot: 1000277705; Cepheid, Sunnyvale, CA, USA), resulting in three inter-run technical replicates. The manufacturer’s recommended PB volume is 4 mL.

Xpert^®^ BCR-ABL Ultra is a single-use, disposable cartridge for use on the GeneXpert^®^ platforms (Cepheid, Sunnyvale, CA, USA) that measures e13a2 and e14a2 *BCR::ABL1* fusion transcripts. The GeneXpert^®^ System (System GX–IV^®^; Cepheid, Sunnyvale, CA, USA) is an advanced instrument designed for molecular diagnostics, featuring single-use cartridges that contain the reverse transcription and PCR reagents and host the reactions. After the PCR reaction, the GeneXpert^®^ software Dx (version 5.1; Cepheid, Sunnyvale, CA, USA) processes the data to report the amount of *BCR::ABL1* transcripts as *BCR::ABL1* to *ABL1* percent ratios on the International Scale (IS) and as a molecular response (MR), defined as the logarithmic reduction from a baseline of 100% IS.

Each Xpert^®^ BCR-ABL Ultra cartridge has two internal quality control systems: the *ABL1* Endogenous Control and the Probe Check Control (PCC). The *ABL1* Endogenous Control normalizes the *BCR::ABL1* target and ensures that an adequate amount of sample is used in the test. The PCC verifies that the reagent has been rehydrated, the PCR tube has been filled, and that all reaction components in the cartridge, including probes and dyes, are present and functioning. Each Xpert^®^ BCR-ABL Ultra kit comes with a certificate of analysis that includes a lot-specific International Scale-Scaling Factor (IS-SF) and an Efficiency ΔCt Value (EΔCt). The IS-SF corrects the assay’s quantitative output to the IS, while the EΔCt represents the amplicon increase per cycle, derived from the slope of the *BCR::ABL1* ΔCt standard curve. This standard curve, which includes four points, is generated from cartridges tested with in-house secondary standards that are calibrated to the WHO International Genetic Reference Panel for *BCR::ABL1* quantification by RT-qPCR. All four laboratories used the same lot (ID 246), which had an EΔCt of 2.09 and an IS-SF of 2.11. The acceptable Ct ranges are between 8 and 18 for *ABL1* and between 8 and 32 for *BCR::ABL1*. The Xpert^®^ BCR-ABL Ultra assay limit of detection (LoD), which is equivalent to the lower limit of quantitation (LLoQ), is 0.003% (MR^4.52^). Any values below LoD/LLoQ are reported as either POSITIVE [below LoD] or NEGATIVE in the GeneXpert^®^ DX software, depending on the reported Ct values for *BCR::ABL1* and *ABL1*. Users can manually calculate the results according to the following formula provided in the instructions:% (IS) = E_ΔCt_^(ΔCt)^ × 100 × Scaling Factor (SF),
where E_ΔCt_ and SF are lot-specific. PCR Efficiency and International Scale-Scaling Factor values are encoded in the cartridge barcode and ΔCt is calculated by subtracting *BCR::ABL1* Ct from *ABL1* Ct (*ABL1* Ct–*BCR::ABL1* Ct).

Factors that may negatively influence the reliability of the results include different anticoagulants, low sample volume, and prolonged sample conservation. For a full description of the system, please refer to the GeneXpert^®^ Dx System Operator Manual or the GeneXpert^®^ Infinity System Operator Manual [14].

### 4.3. Standard Molecular Analysis

Samples collected in each laboratory were tested in duplicate using assays and methods that are routinely employed according to the R.I.L. (Recommendations and Laboratory Indications) published by the Italian national laboratory network for CML.

All laboratories performed RNA extraction using the Maxwell^®^ CSC RNA Blood Kit (Promega Corporation, Madison, WI, USA). Each laboratory employed a different local method for reverse transcription and amplification of the *BCR::ABL1* transcript, following the manufacturer’s recommendations and R.I.L. These local methods are described below:(1)SensiQuant p210 Master Mix (Bioclarma, Turin, Italy) on the 7900HT Fast Real-Time PCR System (Applied Biosystems, Waltham, MA, USA);(2)In- house method that referred to van Dongen et al. [15], with SuperScript™ VILO™ Master Mix and TaqMan™ Fast Universal PCR Master Mix (2X) on QuantStudio 12K Flex Real-Time PCR System (Applied Biosystems, Waltham, MA, USA);(3)ipsogen BCR-ABL1 Mbcr IS-MMR Kits on Rotor-Gene Q (QIAGEN, Hilden, Germany);(4)BCR-ABL P210 ELITe MGB^®^ Kit (ELITechGroup, Turin, Italy) on the 7500 Fast Dx Real-Time PCR Instrument (Applied Biosystems, Waltham, MA, USA).

### 4.4. External Quality Control Material Analysis

We tested two different types of external quality control materials: Thermo Scientific™ AcroMetrix™ BCR-ABL Panel (RUO) and UK NEQAS BCR::ABL1 Major Quantification program samples (EQA program).

The Thermo Scientific™ AcroMetrix™ BCR-ABL Panel (RUO) (REF 956980, AcroMetrix BCR-ABL Panel Kit; Thermo Fisher Scientific, Waltham, MA, USA) is traceable to the World Health Organization’s first International Genetic Reference Panel for the quantification of *BCR::ABL1*. It contains lyophilized cell line material (glass vials), with 1 × 10^6^ cells per vial (~0.300 million cells in 4.5 mL of the final lysate). The panel includes five levels of fusion gene expression. All laboratories tested the same lot (043019). The lyophilized material was analyzed in triplicate using Xpert^®^ BCR-ABL Ultra.

The UK NEQAS BCR::ABL1 Major Quantification program samples (EQA program) consist of two lyophilized samples with *BCR::ABL1*IS levels ranging from 10% to 0.0032%. All laboratories analyzed the same lot (28052021) in quadruplicate using Xpert^®^ BCR-ABL Ultra.

### 4.5. Statistical Analysis

To assess the repeatability of the measurements obtained using both the local methods and the Xpert^®^ system, the coefficient of repeatability, standard deviation (SD), and coefficient of variation (CV) were calculated.

The EUTOS criteria were used to evaluate concordance between the Xpert^®^ BCR-ABL Ultra test and the comparator assays, even though the clinical sample comparisons are likely to include less than the 50 samples recommended for these criteria to apply [16]. Acceptable concordance is defined as the achievement of 2 out of 3 EUTOS criteria [9]. The samples that tested negative were excluded from the concordance analysis.

Bland–-Altman analysis was carried out to determine the bias and the limit of agreement (LoA) between the Xpert^®^ BCR-ABL Ultra test and the local comparators.

The concordance of MR values was assessed using two different approaches. First, we applied the European LeukemiaNet (ELN) recommendations [4] that are routinely adopted by the LabNet network laboratories and defined MR classes through specific upper and lower thresholds [17,18]. For each *BCR::ABL1*%IS value obtained from both the local methods and the Xpert^®^ BCR-ABL Ultra assay, we assigned an MR category. Values were considered concordant when they fell within the same MR class. The MR categories are specified below:*BCR::ABL1*%IS > 0.1% corresponds to <MR^3^;*BCR::ABL1*%IS ≤ 0.1% corresponds to MR^3^;*BCR::ABL1*%IS ≤ 0.01% or an undetectable transcript with >10,000 *ABL1* copies (in the same cDNA volume used for *BCR::ABL1* testing) corresponds to MR^4^;*BCR::ABL1*%IS ≤ 0.0032% or an undetectable transcript with >32,000 *ABL1* copies corresponds to MR^4.5^;*BCR::ABL1*%IS ≤ 0.001% corresponds to MR^5^.

The second concordance analysis was based on a mathematical comparison. For each result obtained with local methods, the MR value was calculated using the formula 2-log10(*BCR::ABL1*%IS), which corresponds to the approach typically used by the Xpert^®^ BCR-ABL Ultra system. This MR value was compared with the MR value reported by Xpert^®^ BCR-ABL Ultra for the same sample. In this analysis, results were defined as concordant when the absolute difference between the two MR values was less than 0.5 log.

## Figures and Tables

**Table 1 ijms-26-08932-t001:** Coefficient of repeatability, standard deviation (SD), mean of *BCR::ABL1*%IS (Mean %IS), and coefficient of variation (CV), by method and molecular response (MR).

Method	MR	Coefficient of Repeatability	SD	Mean %IS	CV
Cepheid	0.0	5028	1814	16,040	0113
Cepheid	1.0	5474	1975	8563	0231
Cepheid	2.0	0472	0170	0458	0372
Cepheid	3.0	0028	0010	0026	0388
Cepheid	4.0	0004	0002	0005	0308
Cepheid	4.5	0000	0000	0000	-
Cepheid	5.0	0000	0000	0000	-
Local	0.0	25,869	9333	50,564	0185
Local	1.0	10,146	3660	38,645	0095
Local	2.0	0157	0057	0321	0177
Local	3.0	0052	0019	0034	0563
Local	4.0	0006	0002	0005	0436
Local	4.5	0002	0001	0001	0732
Local	5.0	0001	0000	0000	2959

**Table 2 ijms-26-08932-t002:** Acceptable concordance according to the EUTOS criteria excluding negative samples.

	% of Samples Achieving the 1st EUTOS Criterion	% of Samples Achieving the 2nd EUTOS Criterion	% of Samples Achieving the 3rd EUTOS Criterion
Overall (107 samples)	57.9	83.2	96.3
Lab #1 (33 samples)	69.7	84.8	100.0
Lab #2 (29 samples)	51.7	82.8	89.7
Lab #3 (23 samples)	60.9	78.3	95.7
Lab #4 (22 samples)	45.5 *	86.4	100.0

* EUTOS criteria non-achievement.

**Table 3 ijms-26-08932-t003:** Bias and 95% limits of agreement (95% LoA) from Bland–Altman analysis, stratified by molecular response (MR).

MR	Bias	95% LoA Lower	95% LoA Upper
MR^1^	19.343	−81.234	119.920
MR^2^	−0.136	−0.799	0.527
MR^3^	0.015	−0.006	0.037
MR^4^	0.003	−0.001	0.006
MR^4.5^ and MR^5^	−0.001	−0.003	0.001

**Table 4 ijms-26-08932-t004:** Comparison of molecular response (MR) values derived from local methods versus the Cepheid method by applying the European LeukemiaNet recommendations that define MR levels by an upper and lower limit.

	<MR^3^ N = 37	MR^3^ N = 29	MR^4^ N = 27	>MR^4.5^ N = 36	Total N = 129
MR local > MR Cepheid	0	0	0	1	1
MR local < MR Cepheid	6	12	18	0	36
N samples concordant Local vs. Cepheid	31	17	9	35	92
% of concordance	83.8	58.6	33.3	97.2	71.3

**Table 5 ijms-26-08932-t005:** Comparison of molecular response (MR) values derived from local methods versus the Cepheid method by applying the formula 2-log_10_(*BCR::ABL1*%IS). The values were defined as concordant if the difference between the two MR values was <0.5 log.

	<MR^3^ N = 37	MR^3–3.99^ N = 29	MR^4–4.49^ N = 27	>MR^4.5^ N = 36	Total N = 129
MR local > MR Cepheid	0	0	0	1	1
MR local < MR Cepheid	2	9	1	0	12
N samples concordant Local vs. Cepheid	35	20	26	35	116
% of concordance	94.6	69	96.3	97.2	89.9

## Data Availability

The data that support the findings of this study are available from the corresponding author upon reasonable request.

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
