# Peer review of "An Automated Cartridge-Based Microfluidic System for Real-Time Quantification of BCR::ABL1 Transcripts in Chronic Myeloid Leukemia: An Italian Experience"

_ijms, 2025, doi:10.3390/ijms26188932_

Round 1
Reviewer 1 Report
Comments and Suggestions for Authors
This study includes a robust multi-center design, thorough comparison with established methods, and clinically relevant validation using reference materials, supporting the potential of Xpert® BCR-ABL Ultra as a standardized tool for CML monitoring.
Despite these strengths, I recommend major revisions to enhance the manuscript’s clarity, methodological consistency, and depth of discussion:
- Explicitly state how this study advances beyond prior validations of Xpert® Ultra (e.g., Dominy et al. 2021) by focusing on multi-center Italian data and real-world applicability.
- Provide a power calculation or rationale for the sample size (n=129), particularly for underrepresented MR4.5/5.0 groups (n=16 and n=20, respectively).
- Address the 10% outlier rate and explain whether these reflect technical variability or systemic biases and discuss implications for clinical use.
- Acknowledge that Xpert®’s inability to report ABL copy numbers prevents strict ELN-defined MR5 categorization, potentially affecting clinical interpretation.
- Include statistical comparisons (e.g., % deviation from expected values) for AcroMetrix™ and UK NEQAS results to strengthen validation claims.
- Clarify whether RNA stability differences between storage methods (RT, 4°C, lysate) influenced concordance at +24h, given the single discordant sample.
- Explain the 36 cases where local MR < Cepheid MR—is this due to Xpert®’s higher sensitivity or methodological differences? Relate this to clinical decision thresholds.
Author Response
We sincerely thank the reviewers for your valuable feedback on our manuscript.
We are grateful for the opportunity to revise our work and provide below a detailed, point-by-point response to the reviewers’ comments. We have taken great care to address all observations and improve the overall quality of the manuscript accordingly.
Reviewer 1 comment 1:
Explicitly state how this study advances beyond prior validations of Xpert® Ultra (e.g., Dominy et al. 2021) by focusing on multi-center Italian data and real-world applicability.
Response:
We addressed this aspect in lines 92–101: “Building on this evidence, our study broadens the validation of Xpert® BCR-ABL Ultra by comparing it across four different methods in a multi-center Italian setting (four GIMEMA LabNet laboratories), using real-world patient samples, external quality controls, and stability testing to reflect routine clinical practice. By focusing on multiple centers and real-world applicability, this work advances beyond prior validations, providing robust evidence of assay performance under different conditions. GIMEMA LabNet ensures that participating centers report results according to the International Scale (IS) through yearly external quality assessment (EQA) rounds, thereby maintaining method standardization. Within this framework, introducing new tools capable of improving laboratory performance remains a key area of research and technical development.”
Reviewer 1 comment 2:
Provide a power calculation or rationale for the sample size (n=129), particularly for underrepresented MR4.5/5.0 groups (n=16 and n=20, respectively).
Response:
We addressed this aspect in lines 127–130: “The number of samples at each molecular response level was inherently limited by the availability of leftover diagnostic material. Notably, samples at MR4.0 and MR4.5 were fewer than other categories, reflecting the lower frequency of patients at these deep molecular response levels encountered in routine laboratory practice.”
Reviewer 1 comment 3:
Address the 10% outlier rate and explain whether these reflect technical variability or systemic biases and discuss implications for clinical use.
Response:
We addressed this aspect in lines 271–277: “This primarily reflects the expected higher variability at low MR levels (MR1 and MR2), where relative measurement differences are inherently larger. Importantly, these levels correspond to high disease burden, and the numerical differences between MR1 and MR2 do not affect clinical decision-making, as they are typically reported together as <MR3 on the patient report. Therefore, these outliers do not indicate a systematic bias of the Xpert® BCR-ABL Ultra assay and are consistent with variability encountered in real-world laboratory settings.”
Reviewer 1 comment 4:
Acknowledge that Xpert®’s inability to report ABL copy numbers prevents strict ELN-defined MR5 categorization, potentially affecting clinical interpretation.
Response:
We addressed this aspect in lines 397-405: “The Xpert® BCR-ABL Ultra does not report the ABL1 copy number in its output; however, it only provides a result if the ABL1 copy number exceeds 32,000. Consequently, if ELN recommendations are strictly applied, no sample analyzed with the Cepheid method can technically be classified as MR5, as the guidelines require more than 100,000 ABL1 copies to confirm this level. It should be noted, however, that the Cepheid method does quantify the ABL1 copy number, and this value directly influences the reported BCR::ABL1%IS and, consequently, the calculated MR value. Therefore, when MR is calcu-lated using the formula implemented by the Xpert® BCR-ABL Ultra, higher MR values can be achieved, and concordance with local methods is substantially improved.”
Reviewer 1 comment 5:
Include statistical comparisons (e.g., % deviation from expected values) for AcroMetrix™ and UK NEQAS results to strengthen validation claims.
Response:
We addressed this aspect in the Supplementary Table S2 and in the manuscript (lines 320–329): “Measurements on AcroMetrix™ reference materials using Xpert® BCR-ABL Ultra showed deviations from expected values ranging from -27.2% to +52.9%. Excellent agreement was observed at clinically relevant low levels (MR4–MR4.5), whereas higher deviations oc-curred at intermediate (MR3) and high (MR1) levels. Measurements on UK NEQAS refer-ence materials using Xpert® BCR-ABL Ultra showed deviations from expected values of -33.7% and -7.5%. The decision to test these two reference materials was driven not only by the need to verify the results obtained from the four different laboratories, but also by the intention to confirm their suitability for use on the Xpert® BCR-ABL Ultra in periodic external quality control rounds that are already being conducted at the international level.”
Reviewer 1 comment 6:
Clarify whether RNA stability differences between storage methods (RT, 4°C, lysate) influenced concordance at +24h, given the single discordant sample.
Response:
We addressed this aspect in lines 411–420: “To simulate real-world conditions, we assessed the stability of test results 24 hours after sample collection. Analysis of 51 samples tested immediately and again after 24 hours demonstrated overall excellent stability. Concordance across different storage conditions—whole blood at room temperature (RT) or 4°C, and lysate at −20°C—indicates that these conditions did not meaningfully affect the results (Supplementary Table S1). The largest deviation was observed in samples stored at RT, which is the condition most prone to RNA degradation. Despite the limited sample size and the need for further confirmation, these results support the robustness of the Xpert® BCR-ABL Ultra assay, making it suitable for integration into typical clinical laboratory workflows, where processing delays beyond 24 hours from collection may occasionally occur.”
Reviewer 1 comment 7:
Explain the 36 cases where local MR < Cepheid MR—is this due to Xpert®’s higher sensitivity or methodological differences? Relate this to clinical decision thresholds.
Response:
We addressed this aspect in lines 375–386: “In these 36 discordant cases, where local MR values were lower than those obtained with the Xpert® BCR-ABL Ultra, the discrepancy is most likely attributable to methodological differences rather than a genuine difference in analytical sensitivity. A key contributor may be differences in ABL1 quantification, which tend to produce systematically higher MR values with the Cepheid method. In some cases, this resulted in samples being classified just above clinically relevant thresholds such as MR3, potentially affecting treatment response categorization under ELN guidelines. This underscores the need for consistent methodology when MR values are near decision boundaries, and for further investigation into the clinical relevance of MR values obtained by applying ELN criteria to Cepheid results. Such an approach differs from the assay’s intended design, which generates MR values through an automated mathematical calculation within the instrument.”

Reviewer 2 Report
Comments and Suggestions for Authors
In this study Danzero and colleagues used the Xpert® BCR-ABL Ultra cartridge to evaluate BCR::ABL1 in samples from either patients affected by CML and reference material, and compared results obtained with the analysis on the same samples performed by four Italian centers belonging to the Italian CML LabNet, observing concordance of the results and thus suggesting the possibility to use the Xpert® BCR-ABL Ultra cartridge in clinical routine to monitor minimal residual disease in CML patients.
Despite this cartridge has already been studied in comparison with Local Diagnostic Assays (LDA, reference 8 cited in the article), I think that the comparison with an Italian setting is relevant. The cohort of patient is well structured and the comparison with data of four centers strenghten results obtained.
As minor revisions, I would suggest:
1) please check gene and transcript format along the manuscript. They should be in Italics.
2) lines 20 and 45: invert the order of genes, the fusion is between BCR on chromosome 22 and ABL1 on chromosome 9 (the same order indicated in transcripts names)
3) lines 53-57: last European LeukemiaNet recommendation (https://doi.org/10.1038/s41375-025-02664-w) suggest a different timeline for molecular monitoring, please refer to the latest literature for this topic
4) line 84: pleaase specify if the stability refers to the test itself or to the results obtained with the test on blood stored for 24h
5) line 104-106: please specify how many samples have been stored in the three methods describe. Please also describe why samples have been stored in three different ways and, if possible, if RNA degradation may have happened considered the storage conditions
6) lines 108-111: it is not fully clear to me the amount of material you have analysed: if the amount of PB is 9-12 ml (3-4 ml for each replicate) or if you obtain three samples from the same 3-4 ml; in the first case, please specify that PB from healthy subjects has been used differently
7) in line 149-150: please insert the reference of the Recommendations and Laboratory Guidelines
8) considering that there are no limits on word count, I suggest to describe results in more details, maybe addinng a figure/table of data not shown
Comments on the Quality of English LanguageThe quality of English language is good on average, the text is easily readable even if sometimes it is a little bit too summarized, that makes comprehensibility a little bit harder.
I only suggest to describe those sections (results and introduction) in more words to make it more easily readable.
Author Response
We are grateful for the opportunity to revise our work and provide below a detailed, point-by-point response to the reviewers’ comments. We have taken great care to address all observations and improve the overall quality of the manuscript accordingly.
Reviewer 2 comment 1:
Please check gene and transcript format along the manuscript. They should be in Italics.
Response:
Thank you for bringing this serious error to our attention. Gene and transcript formatting throughout the manuscript has been corrected to italics.
Reviewer 2 comment 2:
Lines 20 and 45: invert the order of genes, the fusion is between BCR on chromosome 22 and ABL1 on chromosome 9 (the same order indicated in transcripts names).
Response:
We thank you for identifying this important mistake and bringing it to our attention. The order of the genes has been reversed on lines 21 and 48-49.
Reviewer 2 comment 3:
Lines 53-57: last European LeukemiaNet recommendation (https://doi.org/10.1038/s41375-025-02664-w) suggest a different timeline for molecular monitoring, please refer to the latest literature for this topic.
Response:
We appreciate your careful review in pointing out this critical error. The timeline for molecular monitoring has been updated (lines 59–64): “Molecular monitoring is recommended at least every three months until major molecular remission (MMR) is reached and confirmed. Once stable MMR or deeper molecular responses are achieved, testing can be performed at intervals of four to six months. More frequent assessments are warranted if transcript levels show variability or increase, or when evaluating eligibility for treatment discontinuation and during subsequent follow-up in selected patients [3].”
Reviewer 2 comment 4:
Line 84: please specify if the stability refers to the test itself or to the results obtained with the test on blood stored for 24h.
Response:
We appreciate your observation and thank you for bringing this to our attention. The correction has been made on lines 106–108: “To assess its performance, we compared Xpert® BCR-ABL Ultra with established RT-qPCR reference methods using EUTOS criteria [9], evaluated the stability of the results obtained with the assay on blood samples analyzed immediately and after 24 hours from collection, and tested accuracy and reliability using two external quality control materials.”
Reviewer 2 comment 5:
Line 104-106: please specify how many samples have been stored in the three methods describe. Please also describe why samples have been stored in three different ways and, if possible, if RNA degradation may have happened considered the storage conditions.
Response:
Thank you for your valuable comment on this point. The number of samples stored using the three different methods and the rationale for these storage conditions have been added in the manuscript at lines 299–303: “We selected 51 samples, which were analyzed both upon arrival (Time 0) and after 24 hours (Time +24h): 23 samples stored as lysates at −20°C, 14 samples stored as whole blood at room temperature, and 14 samples stored as whole blood at 4°C, to simulate three conditions that may occur in routine workflows of hematology-oncology diagnostic laboratories.”
Considerations regarding potential RNA degradation have been addressed in the Discussion section at lines 411–420: “To simulate real-world conditions, we assessed the stability of test results 24 hours after sample collection. Analysis of 51 samples tested immediately and again after 24 hours demonstrated overall excellent stability. Concordance across different storage conditions—whole blood at RT or 4°C, and lysate at −20°C—indicates that these conditions did not meaningfully affect the results. The largest deviation was observed in samples stored at RT, which is the condition most prone to RNA degradation. Despite the limited sample size and the need for further confirmation, these results support the robustness of the Xpert® BCR-ABL Ultra assay, making it suitable for integration into typical clinical laboratory workflows, where processing delays beyond 24 hours from collection may occasionally occur.”
Reviewer 2 comment 6:
Lines 108-111: it is not fully clear to me the amount of material you have analysed: if the amount of PB is 9-12 ml (3-4 ml for each replicate) or if you obtain three samples from the same 3-4 ml; in the first case, please specify that PB from healthy subjects has been used differently.
Response:
The amount of material used has been clarified in lines 137–139: “Each sample was analyzed in triplicate, with 3–4 mL of the same PB loaded onto each of three Xpert® BCR-ABL Ultra cartridges (Lot: 1000277705), resulting in three inter-run technical replicates. The manufacturer’s recommended PB volume is 4 mL.”
Reviewer 2 comment 7:
In line 149-150: please insert the reference of the Recommendations and Laboratory Guidelines
Response:
It is not possible to provide a reference for this document because it is not publicly available; it is only supplied annually to laboratories participating in the network.
Reviewer 2 comment 8:
Considering that there are no limits on word count, I suggest to describe results in more details, maybe adding a figure/table of data not shown.
Response:
We appreciate your feedback and have carefully considered your suggestion. We have expanded the description of the results and added an additional table presenting the data from the samples analyzed after 24 hours, as suggested.
Reviewer 2 comments on the Quality of English Language:
The quality of English language is good on average, the text is easily readable even if sometimes it is a little bit too summarized, that makes comprehensibility a little bit harder.
I only suggest to describe those sections (results and introduction) in more words to make it more easily readable.
Response: We have expanded all sections of the manuscript, making the description of the work carried out and the results obtained clearer and more specific. In addition, we have reviewed all tables to enhance their readability and clarity. Specifically, formatting was adjusted to better present our results, and the captions were refined to more clearly explain the content of each table.

Round 2
Reviewer 1 Report
Comments and Suggestions for Authors
The authors addressed my comments carefully